# Diagnostic Stewardship—The Impact of Rapid Diagnostic Testing for Paediatric Respiratory Presentations in the Emergency Setting: A Systematic Review

**DOI:** 10.3390/children9081226

**Published:** 2022-08-13

**Authors:** Keshani Weragama, Poonam Mudgil, John Whitehall

**Affiliations:** Department of Paediatrics, School of Medicine, Western Sydney University, Campbelltown, NSW 2560, Australia

**Keywords:** antibiotics, antimicrobial resistance, paediatrics, rapid diagnostic testing, diagnostic antibiotic stewardship, respiratory tract infections, paediatric emergency department

## Abstract

Antimicrobial resistance is a growing public health crisis, propelled by inappropriate antibiotic prescription, in particular the over-prescription of antibiotics, prolonged duration of antibiotic therapy and the overuse of broad-spectrum antibiotics. The paediatric population, in particular, those presenting to emergency settings with respiratory symptoms, have been associated with a high rate of antibiotic prescription rates. Further research has now shown that many of these antibiotic prescriptions may have been avoided, with more targeted diagnostic methods to identify underlying aetiologies. The purpose of this systematic review was to assess the impact of rapid diagnostic testing, for paediatric respiratory presentations in the emergency setting, on antibiotic prescription rates. To review the relevant history, a comprehensive search of Medline, EMBASE and Cochrane Database of Systematic Reviews was performed. Eighteen studies were included in the review, and these studies assessed a variety of rapid diagnostic testing tools and outcome measures. Overall, rapid diagnostic testing was found to be an effective method of diagnostic antibiotic stewardship with great promise in improving antibiotic prescribing behaviours. Further studies are required to evaluate the use of rapid diagnostic testing with other methods of antibiotics stewardship, including clinical decisions aids and to increase the specificity of interventions following diagnosis to further reduce rates of antibiotic prescription.

## 1. Introduction

Antimicrobial resistance (AMR) is a growing and complex public health concern rendering once successful antimicrobial treatments ineffective [1]. Continued AMR, compounded with a reduced number of new microbial agents, can diminish the medical benefits of antimicrobial therapy thus increasing morbidity, mortality, and financial burden within health-care systems [2]. Current evidence highlights antibiotic use and health-care contact as the main drivers in AMR, particularly in multi-resistant bacteria, methicillin-resistant *Staphylococcus aureus*, and antibiotic-resistant *Escherichia coli* [2].

There is an abundance of evidence exploring inappropriate antibiotic prescription, in particular, the use of broad-spectrum antibiotics, over-use of antibiotics, excessive antibiotic treatment duration and unnecessary antibiotic prescription, and its contribution to AMR [1]. These inappropriate antibiotic prescribing behaviours are also associated with increased risk of adverse side effects of antimicrobial agents as well as increased medicalisation of typically self-limiting presentations [3]. In the acute-setting, appropriate selection and administration of antibiotic therapy is often the role of an emergency-department (ED) practitioner, highlighting a potential intervention site for antibiotic stewardship [4]. Changes in practice including the reduction of unnecessary antibiotic prescription for soft tissue and skin infections and viral upper respiratory tract infections (URTIs), as well as encouraging the use of culture and sensitivity data prior to antibiotic prescription have previously been recommended to improve antibiotic prescription behaviours [4].

Over-prescription of antibiotics in paediatric population has been identified as a driver in AMR [5]. Within the paediatric population, factors including parental pressure, medical liability in the context of potentially life-threatening bacterial infections, diagnostic uncertainty and fear of increased morbidity and mortality have been attributed to this pattern of over-prescription [6]. Acute respiratory infections are a leading cause of emergency presentation for the paediatric population [7]. However, it has been found that in children under 5 years of age, up to 66% of cases are due to viral infections [8]. Therefore, antibiotics are being prescribed over double the required amount, encouraging increased antibiotic side-effects and financial burden whilst accelerating AMR [9]. This evidence highlights acute paediatric respiratory presentations in the emergency setting as a potential intervention site for antimicrobial stewardship programs (ASPs).

Current research shows that ASPs can be effective in improving antibiotic prescribing behaviours in the paediatric emergency setting [10]. Several methods of antibiotic stewardship have been explored with rapid diagnostic testing showing great promise however it requires further evaluation to determine its benefits and downfalls in clinical practice [10].

This systematic review aimed to assess the efficacy of rapid diagnostic testing in improving antibiotic prescribing behaviours for respiratory presentations in the paediatric emergency setting. The primary outcome of the study was to assess whether the introduction of rapid diagnostic testing for paediatric respiratory presentations in the emergency setting rendered an improvement in antibiotic prescription, in particular reduced rate of prescription, duration of antibiotic therapy and dosing as well as increased use of narrow spectrum antibiotics. We also assessed whether the implementation of rapid diagnostic testing resulted in improved or comparable clinical outcomes in comparison to standard care.

## 2. Materials and Methods

A systematic review was performed to identify primary journal articles that assess the efficacy and safety of rapid diagnostic testing in managing antimicrobial stewardship, whilst maintaining patient care. The Preferred Reporting Items for Systematic Reviews and Meta-analyses (PRISMA) guidelines were used as a template when performing the review.

### 2.1. Focused Question

Does the implementation of rapid diagnostic tests, for acute respiratory presentations in the paediatric emergency setting, affect prescription rates, duration and dosage of antibiotic prescriptions?

### 2.2. PICO Question

P (population): paediatric patients (aged 1 month–18 years old) presenting to the emergency department with respiratory symptoms;

I (intervention): rapid diagnostic testing;

C (comparison): usual care;

O (outcome): appropriate antibiotic prescribing, whilst maintaining patient care.

### 2.3. Search Strategy

The literature review was conducted on 11 February 2022 across MEDLINE, EMBASE and the Cochrane Database of Systematic Reviews for publications from 1 January 2002 to 11 February 2022, including studies from the last twenty years. The search strategy was discussed with a senior librarian and incorporated specific search terms including ‘paediatric’, ‘rapid diagnostic’, ‘emergency department’ and ‘antibiotics’. The complete search strategy utilised for MEDLINE is included as Table A1. This search strategy was then adapted for the remaining databases. An English-only language filter was used on across all databases.

### 2.4. Eligibility Criteria

#### 2.4.1. Inclusion

Articles were eligible for full-text review if they were primary articles, conducted in the paediatric population presenting to the ED with respiratory symptoms, with clearly defined rapid diagnostic testing methods as the main intervention and when antibiotic prescribing behaviours were at least one of the reported outcomes. Studies comparing two different rapid diagnostic testing methods were also included. Quantitative studies, such as randomised controlled trials, cohort studies and before-and-after studies, were included, as well as relevant systematic reviews for cross-referencing.

#### 2.4.2. Exclusion

Articles were excluded if they were non-primary journal articles or grey literature. Rapid diagnostic testing conducted in paediatric settings other than the ED were excluded. Studies including both adult and paediatric patients were excluded unless paediatric-specific data could be extracted. Qualitative studies were also excluded.

### 2.5. Study Selection

The study selection process was conducted over three rounds. First, duplicate articles were excluded. Title and abstracts of all studies were then screened, and for the relevant studies full-text analysis was performed applying inclusion and exclusion criteria. The study selection process was independently conducted by two authors (KW and PM) followed by a meeting to discuss, and resolve, any discrepancies.

### 2.6. Study Quality and Risk of Bias

The quality of included articles was assessed using the Integrated Quality Criteria for Systematic Review of Multiple Study Designs (ICROMS) tool [11]. Only articles that met the minimum score and mandatory criteria were included in the study.

### 2.7. Data Extraction

Data extraction was conducted by KW from all included studies using a standardised data extraction form, which summarised important study details including authors, publication year, study duration, study location, objectives, interventions, and key findings.

## 3. Results

### 3.1. Search Results

A total of 232 articles were identified across the three databases. Filters limiting the search to studies published in English from 2002 to 2022 were applied. A total of 31 duplicates were removed, leaving a total of 201 articles for title and abstract screening. In total, 159 studies were excluded during this process. Reasons for exclusion included being a non-primary article, not having a clear rapid diagnostic testing intervention, having a non-paediatric population, not addressing respiratory presentations and for being in a non-ED setting. The remaining 42 articles then underwent full-text analysis from which 18 were eligible for final analysis (Figure 1). The data extracted from these 18 eligible studies are presented in Table 1.

### 3.2. Included Studies

Included studies were published between 2002 [28] and 2021 [13,25]. Eight of eighteen studies were randomised control trials [17,18,19,20,22,23,24,25], three were prospective cohort studies [14,15,26], five were retrospective cohort studies [8,12,21,27,28], one was a before and after interventional study [16], and one was a pragmatic AB single subject study [13]. The studies were conducted in several different countries, the majority being in the United States [8,12,16,18,20,22,24,25,26,27,28], but also other countries including United Kingdom [13], Canada [17], Australia [21], Belgium [14], Brazil [15], Italy [19] and Turkey [23]. All included studies were conducted in the ED of a paediatric hospital or within the paediatric population of a general hospital, however, the number of hospitals varied between studies. Fifteen of eighteen studies reported outcomes from the paediatric population only [8,12,13,15,16,17,19,20,21,23,24,25,26,27,28], and three studies extracted paediatric data from a general hospital [14,18,22].

### 3.3. Intervention

The rapid diagnostic testing intervention varied between included studies. Three studies used rapid streptococcal tests, to assess for the presence of group A streptococcal (GAS) [12,13,15]. Cardoso et al. conducted a prospective study in which they asked clinicians about the patient management plan at two separate time points after ED admission: after clinical examination and then after a rapid streptococcal detection test [15]. Nine studies assessed rapid diagnostic tests which assess for the presence of viruses. These tests included Alere i, Quickvue Influenza Test, direct immuno-fluorescence assay, Influenza A/B Rapid Test as well as the use of rapid PCR and enzyme-linked immunosorbent assay (ELISA) testing for influenza and respiratory syncytial virus (RSV) [14,17,19,20,21,23,24,26,28]. The remaining six studies used the BioFire FilmArray Respiratory Panel 2 (BioRP2). This test can target 18 bacterial species and 9 viruses that cause lower respiratory tract infections [29]. Bird et al. combined the use of a rapid GAS detection test with McIsaac’s clinical score as the intervention [13]. Similarly, Crook et al. also combined the use of a standardised clinical guideline with BioRP2 but were able to pool the data to measure the effects of these components separately [16].

### 3.4. Primary Outcome

The primary outcomes varied across the included studies, ranging from antibiotic prescription rate to length of patient admission. All studies reported changes in antibiotic prescription rate before and after intervention [8,12,13,14,15,16,17,18,19,20,21,22,23,24,25,26,27,28]. Antiviral prescription rate, hospital admission rate, length of stay and use of ancillary testing were also reported in most studies [8,14,16,17,18,19,20,21,22,24,25,26,27,28]. Duration of antibiotic therapy was reported in three studies [8,16,27]. Most studies did not specifically report on clinical outcomes of the patient; however, Iyer et al. and Rao et al. reported the number of patients who returned to ED after discharge [20,25]. Further, Bird et al. assessed the diagnostic accuracy of the RDT intervention and Ayanruoh et al. reported a comparison of culture results against the RDT [12,13]. Rogers et al. reported on the time taken to result in comparison to standard care [27]. Finally, Busson et al. and Sharma et al. also reported on costs to ED [14,28].

### 3.5. Efficacy of RDTs in Antibiotic Stewardship

#### 3.5.1. Group A Streptococcal

All studies utilising a RDT for GAS specifically observed a significant reduction in antibiotic prescription rates.

Two studies assessed the use of GAS RDTs alone. Ayanruoh et al. found a reduction in antibiotic prescription rates for children with symptoms suggestive of pharyngitis, from 41.38% in the pre-RDT period to 22.45% after the introduction of GAS RDT (OR 0.27; 95% CI, 0.24–0.30; *p* < 0.001). Further, the accuracy of the RDT was assessed by obtaining throat cultures of all negative RDT results, and a very low false-negative rate of 0.04% was observed [12]. Cardoso et al. compared the intention of antibiotic prescription by a physician based on clinical findings to the results from a GAS RDT [15]. It was observed that 32.9% of cases would have been prescribed unnecessary antibiotic based on clinical assessment alone. It was also found that 17.1% of suspected cases would not have received the necessary antibiotics for pharyngotonsillitis, putting these patients at risk of adverse complications [15].

Bird et al. combined the use of a GAS RCT with McIsaac’s Clinical Score and assessed its efficacy in reducing antibiotic prescription rates. During the pre-intervention period, the baseline antibiotic prescription rate was 79%. After the introduction of RDTs and McIsaac’s score, supplemented with the use of throat swabs in all negative results, these rates dropped to 24%. In the third phase of the study, supplemental testing was removed and the assessment was based on the intervention alone, and antibiotic prescription rates increased to 28% [13]. Although the study reported poor sensitivity of the RDT and poor specificity of McIsaac’s score, when used in combination there was a reduction in antibiotic prescription rates in the paediatric emergency setting.

#### 3.5.2. Viral Testing

Of the eight studies assess the use of RDT testing for viruses only, three studies utilised the QuickVue Influenza Test. Esposito et al. found that a positive QuickVue test result showed a significant reduction in antibiotic prescription in comparison to cases with a negative result or no testing at all (32.6% vs. 64.8% and 61.8%; *p* < 0.0001 and *p* = 0.0003) [19]. In comparison, Poehling et al., despite finding the QuickVue RDT to be 82% sensitive and 99% specific, found no different in antibiotic prescription rates after the introduction of the intervention into practice (39% vs. 51%, *p* = 0.03) [23]. It was, however, found that fewer children that underwent RDT had further diagnostic testing in comparison to children who were not tested [24]. Iyer et al. compared the use of QuickVue RDT to standard laboratory testing. This study found that a positive test for influenza was associated with a reduction in antibiotic prescription; however, there was no significant difference in these rates between the method of testing (RDT vs. standard laboratory testing) [20].

Busson et al. evaluated the Alere i Influenza A&B test and its ability to contributed to patient management. The utilisation of Alere i RDT in the emergency setting lead to a 42.9% reduction in antibiotic prescription [14]. Ozkaya et al. found that patients who underwent rapid diagnostic testing, using the Influenza A/B Rapid Test were less likely to be prescribed antibiotics in comparison to clinical assessment alone (32% vs. 100%, respectively, *p* < 0.0001) [23]. Rogan et al. assessed the difference in clinical decisions, including antibiotic and antiviral prescription, in standard management and management adjusted for bedside molecular RDT results. Physicians reported they would have reduced antibiotic prescription rates by 17% (*p* = < 0.001) and increase appropriate antiviral prescription by 13% (*p* = 0.023) [26]. Sharma et al. compared antibiotic prescription rates in patients who had a rapid influenza diagnosis prior to discharge (early diagnosis) compared to those who received their results after discharge (late diagnosis) [28]. A significant reduction in dispensing of ceftriaxone sodium from 24% to 2% (*p* = 0.006) was found in the early diagnosis group in comparison to the late diagnosis group [28]. Doan et al. found no significant difference in antibiotic prescription rates following the use of rapid direct immunofluorescence assay in comparison to standard care (RR): 0.86, 95% CI (0.48, 1.53); however, children who underwent RDT were less likely to receive antibiotics from their general practitioner within one week of discharge from emergency care (RR): 0.36, 95% CI (0.14, 0.95) [17]. Finally, Jacob et al. was assessing the effect of RDT location on treatment time, use of ancillary testing and antibiotic prescription in the emergency setting. This study compared the use of RDT at the bedside, RDT use in the laboratory and no RDT. It was found that antibiotic prescription rates highest in patients who did not undergo RDT; however, RDT location alone did not significantly affect these rates. (15.2% vs. 2.7% in the laboratory group and 11.2% in the bedside RDT group; *p* < 0.0001) [21].

#### 3.5.3. Respiratory Panel Testing

Six included studies assessed the efficacy of a BioFire FilmArray Respiratory Panel 2 (BioRP2) as a diagnostic tool in antibiotic stewardship. Three of these studies found a significant improvement in antibiotic prescribing methods after intervention was introduced.

Echavarria et al. compared the use of BioRP2 and immunofluorescence assay (IFA) in the emergency setting [18]. Diagnosis using the BioRP2 resulted in a significant reduction in antibiotic prescription in children (OR: 12.23, 95% CI 1.56–96.06), as well as a reduction in the use of ancillary testing (OR): 9.64, 95% CI 2.13–43.63) in comparison to IFA [18]. It was also noted that the median time from sample collection to result was 1 h 52 min with the BioRP2 in comparison to 26 h by IFA (*p* < 0.001) [18]. Zhu et al. compared the use of BioRP2 in the inpatient and emergency settings [8]. In the emergency setting, patients with a positive BioRP2 result received few antibiotic prescriptions in comparison to patients not tested (−32.3%; *p* < 0.001) [8]. BioRP2 use was, however, more prevalent in the inpatient setting in comparison to ED (78.9% vs. 7.3%; *p* < 0.001) [8]. Rogers et al. compared the use of BioRP2 to standard testing in the form of PCR [28]. This study found no difference in decision to prescribe antibiotics; however, the duration of antibiotic therapy was shorter in those who underwent RDT with the BioRP2 (*p* = 0.003) due to the reduced time to result [27]. May et al. compared the use of this RDT to usual care, with the primary outcome measure being antibiotic prescription [22]. This study found a 12% reduction (95% CI −25–0.4%; *p* = 0.06/0.08) in antibiotic prescription in the RDT arm of the study with a greater reduction seen in paediatric patients (−19%) in comparison to adults (−9%) [22]. Rao et al. conducted a study to assess whether of BioRP2 reduced antibiotic use and further ancillary testing in children presenting to the ED with respiratory symptoms [25]. This study found no significant difference in antibiotic prescribing after the introduction of BioRP2 (RR: 1.1, 95% CI 0.9–1.4); however, it did find that children were more likely to receive antivirals (RR: 2.6, 95% CI 1.6–4.5) [25]. Therefore, this study concluded there was a limited role for RDT for children in the emergency setting [25]. Finally, Crook et al. assessed changes in patient management after the introduction of a clinical guideline and then after adding a RDT in practice [16]. The introduction of the RDT was associated with a reduction of antibiotic prescription of −10.8% (95% CI 6.5–15%), as well as a reduction in the duration of antibiotic therapy by 0.47 days (95% CI 0.16–0.51); however, when adjusted for temporal trends, only the only significant change after the introduction of RDT was the reduction in ancillary testing [16].

### 3.6. Risk of Bias

All papers included in this study were assessed for bias using the ICROMS tool. All studies met the minimum score and respective mandatory criteria based on study design, and were, therefore, deemed to be of good quality and included in the review (Table A2, Table A3 and Table A4).

## 4. Discussion

Antibiotic prescribing behaviours have been described as suboptimal in the emergency setting [30]. Inappropriate antibiotic prescription includes the dispensing of unnecessary antibiotics, the use of broad-spectrum as opposed to narrow spectrum antibiotics, prolonged antibiotic therapy, as well as a lack of patient education [1]. With the growing risk of AMR, antibiotic stewardship practices need to be placed in target locations and populations to reduce the rates of inappropriate antibiotic prescription. The paediatric population, in particular children with respiratory presentations, are target population due to reported overuse of antibiotics [5,31]. There is a great body of evidence showing that febrile infants who are older than 3 months and have a confirmed viral infection, are at very low risk of severe bacterial infection [32]. Therefore, it can be inferred that a rapid confirmation of a viral or non-bacterial diagnosis would reduce the need for antibiotic prescription as well as the use of ancillary testing. Previous studies have shown that the presence of rapid respiratory pathogen testing equipment show great promise in increasing rates of appropriate antibiotic prescription [10]. This systematic review aimed at assessing the efficacy of these rapid diagnostic tests for paediatric patients with respiratory presentations in the emergency setting.

### 4.1. Antibiotic Prescription Behaviours

Decisions to change clinical management of patients, in particular, the decision to prescribe or not prescribe antibiotics, depends on several factors, including the identification of the pathogen and the patient’s medical history [33]. Of the studies included in this review, twelve found a significant decrease in antibiotic prescription rates following the introduction of RDT in clinical practice [8,12,13,14,15,16,18,19,21,23,26,28]. This decrease in antibiotic prescription rates was observed across GAS-only, viral-only, as well as the respiratory panel RDTs review; therefore, it is difficult to ascertain why some studies did not find a significant difference. Poehling et al. mentioned parental anxiety as a factor that should have been measured outcome in the study as it may have affected prescription rates. Interestingly, 20% of patients who tested positive for influenza in both arms of this study still received antibiotics. This is not in line with current literature which shows a significant decrease in antibiotic prescription and increase in the prescription of antivirals, such as oseltamivir in patients with a confirmed diagnosis of influenza, highlighting the need for further investigation into potential barriers [19,34]. Current evidence showcases the positive impact of oseltamivir on patient outcomes [35]. The introduction of RDT was found to improve rates of antiviral prescription in four included studies, in particular Rogan et al. found a 13% increase in antiviral prescription following introduction of diagnostic RDT [26]. Duration of antibiotic therapy is another important measure of appropriate antibiotic prescription. Rogers et al. did not find a significant difference in the rate of antibiotic prescription however there was significant decrease in the duration of antibiotic therapy [27]. This change was attributed to receiving test results within four hours of presentation [27]. Similarly, previous studies have found that in the adult population the introduction of RDT in comparison to standard PCR testing resulted in a decrease in antibiotic therapy duration rather than in the rate of antibiotic prescription. This change was accredited to the reduced need for empirical antibiotic therapy due to the rapid availability of results, decreasing the patient’s overall antibiotic dosage [33].

### 4.2. Patient Outcomes

The success of an antibiotic stewardship intervention, which affects clinical management, cannot be appreciated without assessing clinical outcomes of the patients included in the study population. Of all included studies, two included studies observed whether the patient returned to the ED for the same symptoms following discharge, with both reporting no significant differences between groups [20,25]. Unfortunately, only one study observed 30-day mortality rates of included patients, but found no significant difference after the introduction of RDT in practice [16].

Diagnosis of bacterial pharyngitis, based on clinical assessment using signs and symptoms alone can be a poor diagnostic tool [15]. Typically, children over than five years of age with bacterial disease show signs of fever, headache, vomiting, abdominal pain, purulent tonsillar discharge and hypertrophy, as well as painful anterior cervical lymphadenopathy. Further, the absence of URTI symptoms including a runny nose, cough, watery eyes and diarrhoea contribute to the overall clinical presentation [15]. Therefore, incorporation of a clinical scoring system (i.e., McIsaac Clinical Score), as seen by Bird et al., could be favourable in the diagnostic method [13]. However, clinical assessment should ideally be accompanied with microbiological testing to ensure accurate identification of the pathogen and, thus, management of the patient [15]. This was highlighted in the study conducted by Cardoso et al. in which 17.1% of GAS pharyngotonsillitis cases would not have received the required antibiotics if diagnosis was conducted by clinical assessment alone, placing these patients at risk of complication [15].

Crook et al. found that the introduction of RDT into diagnostic clinical practice has the most significant clinical impact on infants aged 29–60 days old [16]. Typically, infants < 28 days old undergo a comprehensive sepsis screen when presenting to ED with suspected bacterial illness, and infants > 60 days old receive a less comprehensive screen [16]. However, management of infants aged 26–60 days varied greatly between physicians due to the lack of clinical guidance, highlighting a potentially high-risk group for the misuse of antibiotics [16]. Importantly, the decrease in ancillary testing and antibiotic prescription following RDT intervention in this study was not associated with a compromise in clinical care, with the proportion of infants being discharged with a bacterial infection remaining low (<1%) in both groups [16]. Although the flow-on effect of reduced antibiotic prescription rates, including reduced selection of drug-resistant pathogens as well as the maintenance of infant microbiome ensuring advantageous health outcomes, the low rate of patient outcome reporting in the included studies highlights a shortcoming in current literature. This is of concern, as the absence of a significant difference in patient outcomes between groups assessed with RDT, as opposed to standard measures, does not represent non-inferiority. Therefore, further research needs to assess the implementation of RDTs powered for patient outcome measures, rather than antibiotic prescription rates alone. This will show RDTs efficacy for AMR as well as clinical outcomes, prior to implementation in emergency settings.

### 4.3. Acceptability of RDT by Hospital Staff & Patients

Another important factor to consider when implementing an intervention that affects clinical workflow and patient management is the acceptability of said intervention among hospital staff and patients. Clinicians interviewed about the effect of RDT on patient management expressed concern about using RDT results alone to make clinical decisions and stated that clinical assessment as well as results from further ancillary testing would be required to modify patient management [14]. This is in line with current guidelines in the United States which do not solely depend on RDT results, with clinicians required to confirm negative RDT results with a throat culture [12]. It should also be considered that a reduction in ancillary testing, in particular chest X-rays, has been a priority to minimise the adverse effects of testing including unnecessary radiation [36]. Among the nursing staff, who were administering the RDT, there were complaints about the RDT result being too faint to read [13]. Further, one study reported the importance of maintaining adequate technique when obtaining throat swabs for the RDT sample [13]. These potential issues may be reduced by providing adequate training to the staff, and then allowing only trained, permanent nursing staff to administer the RDT. When assessing the acceptability of RDTs in paediatric patients, it is often the parent’s opinion which is considered. Parental expectation of antibiotic prescription is a point of intervention for reducing rates of antibiotic prescription. Clinicians found that having a negative RDT result to show parents empowered them to tell concerned parents that antibiotics were not indicated [13].

### 4.4. Implementing RDTs in the Emergency Setting

Several factors need to be considered when implementing RDTs into clinical workflow, including the accuracy of the diagnosis, financial burden as well as the improvements with regards to time to result.

Previous studies have shown a mixed review of GAS RDTs, with a majority reporting poor sensitivity in detecting GAS pharyngitis [37]. Reported findings from included studies were in line with these findings, reporting a poor sensitivity of 64.3%, but a high negative predictive value of 92.1% in comparison to throat culture [13]. This lack of sensitivity was partially attributed to a deficit in swab technique in staff administering the RDT [13]. Further, it was found that in McIsaac’s Clinical Score had a low specificity of 12.62%, but a high sensitivity of 92.11% [13]. This shows great promise for the combination of RDTs with clinical scoring systems in the diagnosis of GAS pharyngitis, which is in line with current literature that shows an increase of RDT sensitivity from 95.8% to 97% after the introduction of a clinical criteria [38]. The Alere i RDT use for the detection of influenza, reported an adequate sensitivity and specificity of 91.4% and 97.6%, respectively, for Influenza A, and 54.5% and 98.8%, respectively, for Influenza B [14]. Finally, the BioRP2 RDT is advertised for having a good sensitivity of 85–100% and a specificity of 90–100% [22]. From these measures the implementation of RDTs either alone or supplemented by a clinical criterion would most likely benefit clinical diagnosis.

The financial burden of antibiotic over-prescription and the flow-on costs stemming from AMR need to be balanced with the introduction of RDTs into clinical practice. The administration of RDTs to all patients in the ED would not be practical financially or technically due to the high cost of each test and the requirement for extra staff and training for RDT administration [14]. However, several studies highlight the potential of RDT for decreasing costs by reducing hospitalisation days, use of ancillary testing as well as antibiotic prescriptions [14,19]. In order to balance cost and benefit of RDTs, several studies suggested the use of RDTs in only high-risk populations or only during peak influenza seasons to ensure accurate diagnosis and reduce antibiotic use [14,19].

The importance of time taken to receive a result is emphasised due to the common practice of dispensing empirical antibiotics in ill patients. It has been observed that patients who receive antibiotics on discharge prior to result availability, were often not informed that antibiotics could be discontinued after negative results were received [14]. Further, Echavarria et al. found that the discrepancy in time to RDT results which took on average 1 h 52 min in comparison to 26 h for immunofluorescence assays in the laboratory, was associated with a significant reduction in antibiotic prescription rates, as well as further ancillary testing and an increase in appropriate antiviral prescription rates [18]. These results highlight the speed of RDT results as a distinguishing factor that contribute to its success in clinical practice and therefore in reducing the rate of AMR. It should be noted that the waiting time for the rapid test results may delay optimal treatment time; therefore, the dispensing of empirical antibiotics may still be required. Moreover, the ability for RDTs to provide sufficient diagnosis in an emergency setting should be explored. Currently, the BioRP2 RDT may be considered in an emergency as it has the ability to provide a full panel of different pathogens within 45 min [22].

### 4.5. Strengths & Limitations

This study has many strengths. The inclusion of only those studies that met the mandatory criteria from the ICROMS risk of bias assessment tool ensured that only high-quality studies were reviewed. Furthermore, a variety of RDT interventions, testing for different pathogens, were included, providing a generalisable overview of this diagnostic method. This provides clinicians and hospital staff ample information for the development and implementation of RDTs in the future.

This study also had some limitations. Only three databases were included in the initial search; therefore, some references may have been overlooked. Further, only papers published in English were included, potentially overlooking relevant papers published in other languages. Moreover, the inclusion of a variety of RDTs, assessing multiple pathogens, may have led to an inconclusive overview of each rather than a comprehensive study assessing one type of RDT.

## 5. Conclusions

We conclude that RDTs are an effective method of diagnostic antibiotic stewardship whilst aiding with appropriate diagnosis and therefore management of paediatric patients with respiratory presentations in the emergency setting. Successful administration of RDTs have shown great promise in improving antibiotic prescribing behaviours by reducing antibiotic prescription rates, duration of antibiotic therapy and promoting the use of appropriate antiviral therapy. Further research is required to assess the effect of RDT implementation with a focus on patient clinical outcomes. We recommend further research into the effectiveness of RDTs supplemented with clinical decision aids and criterion, to increase the specificity of the intervention to ensure appropriate treatment of patients and further reduce unnecessary antibiotic prescription rates.

## Figures and Tables

**Figure 1 children-09-01226-f001:**
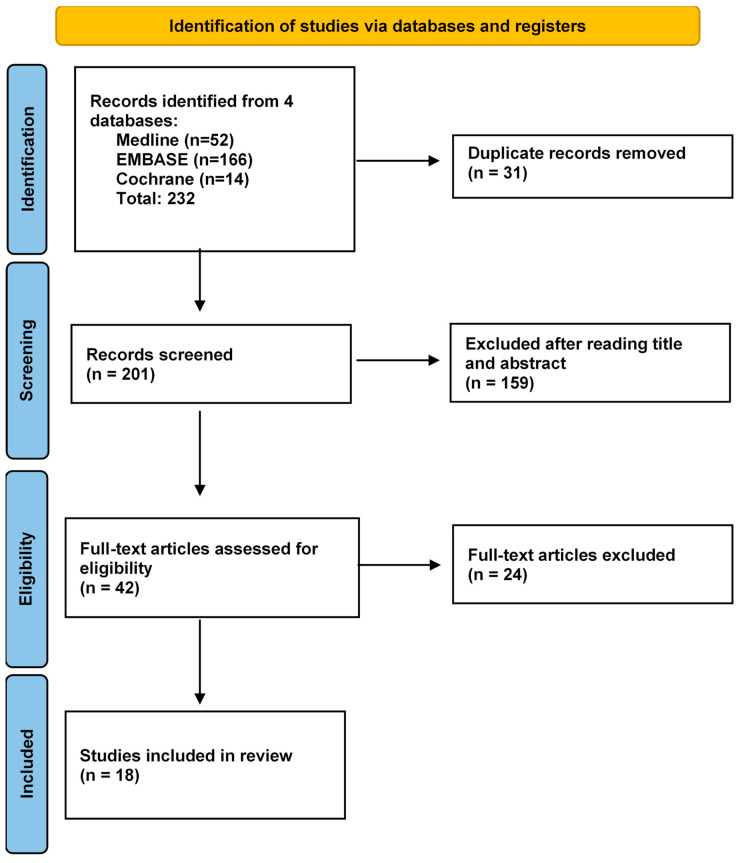
Preferred Reporting Items for Systematic Reviews and Meta-Analysis (PRISMA) Flow Diagram of Study Selection Process.

**Table 1 children-09-01226-t001:** Diagnostic Stewardship–The Impact of Rapid Diagnostic Testing for Paediatric Respiratory Presentations in the Emergency Setting: A Systematic Review.

Author; Year, Country; Study Period; Setting	Study Design; Population and Sample Size	Objective	Intervention	Key Findings
Ayanruoh et al., 2009 [12]; USA; September 2005–September 2007; Paediatric ED of inner-city hospital in New York City	Retrospective Cohort; patients aged 3–18 years old, n = 8280	Assess the impact of RDTs on antibiotic prescriptions in children with pharyngitis in the ED	Rapid Streptococcal Test for GAS	Introduction of RDTs was associated with a lower antibiotic prescription rate from 41.38% pre-RDT and 22.45% post-RDT, *p* < 0.001.
Bird et al., 2021 [13]; UK; 2014–2016; Birmingham Children’s Hospital ED	Pragmatic AB single-subject study; children aged 6 months to 16 years presenting to ED with a sore throat, n = 605	Assess efficacy of RDT for GAS combined with established clinical scoring system (McIsaacs Score) in reducing antibiotic prescribing for sore throat	McIsaac clinical score combined with GAS RDT to screen for and treat GAS pharyngitis	Baseline rates of antibiotic prescription rate 79% was reduced to 24% following the implementation of intervention.
Busson et al., 2017 [14]; UK; 2 January–30 March 2015; Saint Pierre’s University Hospital	Cohort study; paediatric patients (<15 years old) presenting to ED with fever of unknown source, suspected influenza, or complicated illness	Evaluate the contribution of the Alere i Influenza A&B test to patient management	Alere i Influenza A&B test	Antibiotics were avoided in 36.2% of patients (20 paediatric patients, 5 adult patients). Antivirals prescribed in 36.2% patients (9 paediatric patients, 16 adult patients).
Cardoso et al., 2013 [15]; Brazil; April-November 2008; University Hospital of Sao Paulo	Cohort study; patients aged between 2 and 15 years with history of sore throat and fever, no signs of viral infection, n = 650	Evaluate the impact of RDTs on the diagnosis and treatment of children with acute pharyngotonsillitis	Clearview Strep A Test (Oxoid), which consists of a rapid immunoassay for the qualitative detection of group A streptococcal antigens (RADT)	Implementation of RDT prevented unnecessary antibiotic prescription in 32.9% of cases.
Crook et al., 2020 [16]; USA; 3 study periods between January 2011 and June 2019; Tertiary-care children’s hospital	Before-and-after intervention study; children aged < 90 days presenting to ED with fever or hypothermia, n = 5317	Assess the impact of clinical guidelines and RDTs on paediatric patient management	BioFire FilmArray Respiratory Panel 2 & mPCR testing with clinical guidelines	Introduction of RDT was associated with a significant reduction in antibiotic prescription in children aged 29–60 days old.
Doan et al., 2009 [17]; Canada; December 2005–April 2006 and November 2006–April 2007; British Columbia Children’s Hospital	RCT; children 3 to 36 months of age with febrile acute respiratory tract infections at a paediatric ED, n = 204	Assess whether early and rapid diagnosis of a viral infection alleviates the need for ancillary testing and antibiotic treatment	Rapid respiratory viral testing program, named VIRAP (for Viral Rapid Program)	No statistically significant difference in ED length of visits, rate of ancillary testing, or antibiotic prescription rate in the ED between the study groups. Significant reduction in antibiotic prescription after ED discharge (in the group who had rapid viral testing RR = 0.36; 95% CI = 0.14, 0.95).
Echavarria et al., 2018 [18]; Argentina; April-November 2016 and April-October 2017; University Hospital, Buenos Aires, Argentina	Prospective, randomized, non-blinded study; patients aged 2 months–6 years of age (children) or greater than 18 years (adults), with signs/symptoms of acute lower respiratory infection with onset within the preceding 7 days	Determine if timely etiological diagnosis has an impact on medical management in relation to antibiotic and antiviral prescription, and use of complementary studies	BioFire FilmArray Respiratory Panel 2 or immunofluorescence assay (IFA)	Diagnosis with FilmArray-RP was associated with significant changes in medical management including withholding antibiotic prescriptions in children (OR: 12.23, 95% CI: 1.56–96.09).
Esposito et al., 2003 [19]; Italy; 6 January–27 February 2002; University of Milan, Italy	RCT; patients aged 0–15 years attending ED because of influenza-like illness, n = 957	Assess the effect of a rapid diagnosis of influenza infection on the management of children with influenza-like illness in an ED	Quickvue Influenza Test	Patients with a positive Quickvue test were significantly less likely than those with a negative or no test result to receive antibiotics (32.6% vs. 64.8% and 61.8%; *p* < 0.0001 and *p* = 0.0003).
Iyer et al., 2006 [20]; USA; 27 January 2003–31 March 2003 and 8 December 2003–29 January 2004 pediatric ED of a large, urban, tertiary-care pediatric teaching hospital	Prospective, quasi-RCT; febrile children at risk for serious bacterial illness (SBI) based on age and temperature and who presented to a paediatric ED during an influenza outbreak, n = 700	To determine the effect of point-of-care testing (POCT) for influenza on the physician management of febrile children	Quickvue Influenza Test	No significant differences were demonstrated between the POCT and ST groups with respect to laboratory tests ordered, chest radiographs obtained, antibiotic administration, inpatient admission, return visits to the pediatric ED, lengths of stay, or visit-associated costs.
Jacob et al., 2021 [21]; Australia; 1 August 2017–30 September 2017; ED, The Children’s Hospital at Westmead	Retrospective, observational study; patients included were aged < 16 years who had Influenza-like illness, n = 1451	Assess whether location of rapid influenza diagnostic testing (RIDT) for patients with influenza-like illness has an impact on ED treatment time or ancillary testing	BD Veritor digital immunoassay (bedside) or Quidel Sophia fluorescence immunoassay (laboratory)	Location of RIDT may not have a significant impact on treatment time, ancillary testing and treatment with antibiotics. When RIDT was not performed, patients had the shortest treatment time.
May et al., 2019 [22]; USA; December 2016-over 2 winter respiratory seasons and 1 intervening non-respiratory season; level 1 emergency department	Prospective, patient-oriented, pilot RCT; patients ≥ 12 months old, had symptoms of upper respiratory infection or influenza- like illness, and were not on antibiotics, n = 191	Evaluate whether having a RDT result available during the ED visit would have a significant impact on management and outcomes in patients with clinical signs and symptoms of acute respiratory tract infection	BioFire FilmArray Respiratory Panel 2	Twenty (22%) RDT patients and 33 (34%) usual care patients received antibiotics during the ED visit (–12%; 95% confidence interval, –25% to 0.4%; *p* = 0.06/0.08); 9 RP patients received antibiotics despite having a virus detected. The magnitude of antibiotic reduction was greater in children (–19%) vs. adults (–9%, post hoc analysis). There was no difference in antiviral use, length of stay, or disposition.
Ozkaya et al., 2009 [23]; Turkey; November 2006 and March 2007; Vakıf Gureba Education and Research Hospital, Istanbul, Turkey	RCT; Patients aged 3 to 14° years presenting to ED with fever and cough, coryza, myalgias and/or malaise, n = 97	Determine the influence of rapid diagnosis of influenza on antibiotic prescribing to children presenting with influenza-like illness in the ED	Influenza A/B Rapid Test	Patients in RDT group were less likely to be prescribed antibiotics when compared to those in usual care (32% vs. 100%, respectively, *p* < 0.0001).
Poehling et al., 2006 [24]; USA; 28 January–8 April 2003 and 1 December 2003 to 31 January 2004; Vanderbilt Pediatric ED	RCT; patients < 5 years of age presenting with any of the following symptoms: cough, rhinorrhoea, wheezing, difficulty breathing, fever, sore throat, apnoea, or ear pain, n = 468	Determine whether a point-of-care rapid influenza test impacts the diagnostic evaluation and treatment of children with acute respiratory illnesses	PCR + RDT (QuickVue Influenza Test)	In the ED, fewer children in the rapid test group had diagnostic tests ordered than in the no rapid test group (39% vs. 51%, *p* = 0.03). There was no difference in test ordering in the clinic or in antibiotic prescribing in either setting.
Rao et al., 2021 [25]; USA; 1 December 2018–30 November 2019; The Children’s Hospital Colorado	RCT; children aged 1 month to 18 years presenting to an ED with ILI, n = 931	Determine whether RDT testing leads to decreased antibiotic use and healthcare use among children with influenza-like illness in an ED	BioFire FilmArray RP2 Panel	The use of RDT testing in the ED for ILI did not decrease antibiotic prescribing in this randomized clinical trial (RR, 1.1; 95% CI, 0.9–1.4). There is a limited role for RRP pathogen testing in children in this setting.
Rogan et al., 2017 [26]; USA; 10 January 2016–13 March 2016; Pediatric ED at Stanford University Medical Centre	NCBA; consecutive paediatric patients <18 years of age who had a respiratory virus PCR panel by nasopharyngeal swab, n = 28	Determine the impact of bedside PCR on paediatric acute respiratory infection management	PCR test for respiratory viruses	Physicians would have decreased ED LOS by 33 min, ordered fewer tests (18%; *p* < 0.001) with average patient charge savings of $669, fewer antibiotics among discharged patients (17%; *p* < 0.043), and increased appropriate antiviral use (13%; *p* < 0.023).
Rogers et al., 2015 [27]; 1 November 2011–31 January 2012, and 1 November 2012–31 January 2013; Children’s Healthcare of Atlanta	NCBA; patients who aged 3 months to 21 years who had respiratory panel test, n = 771	Determine if implementation of the RDT led to a shorter time to the test result and expanded panel, results in different outcomes for children admitted to the hospital with an acute respiratory tract illness	BioFire FilmArray RP2 Panel	The RRP decreases the duration of antibiotic use (*p* = 0.003), the length of inpatient stay (*p* = 0.03), and the time in isolation (*p* = 0.03).
Sharma et al., 2002 [28]; USA; 1 November 1998, through 30 April 2000; urban children’s hospital ED	Retrospective cohort: all children 2 to 24 months of age, with a temperature higher than 39 °C who had a positive influenza virus type A test result using an enzyme-linked immunosorbent assay, n = 72	Determine the effect of rapid diagnosis of influenza virus type A on the clinical management of febrile infants and toddlers in a paediatric ED	Rapid detection of influenza virus type A infection by enzyme-linked immunosorbent assay (ELISA)	Fewer patients in the early diagnosis group received ceftriaxone sodium compared with those in the late diagnosis group (2% vs. 24%, *p* = 0.006); there were fewer urinalyses (2% vs. 24%, *p* = 0.006) and complete blood cell counts performed (17% vs. 44%, *p* = 0.02).
Zhu et al., 2019 [8]; USA; 16 December 2013–15 December 2015; ProMedica Toledo Children’s Hospital	Retrospective cohort: children 1 month to 18 years of age with uncomplicated acute respiratory tract infections admitted into the hospital or seen in the ED, n = 939	Assess whether RPP decreases antibiotic days of therapy and length of hospital stay for paediatric patients with acute respiratory infections	BioFire FilmArray RP2 Panel	Fewer RPP-positive patients were prescribed antibiotics on discharge when compared with RPP-negative patients (8.8% vs. 41.1%; χ2 = 13.57; *p* < 0.001). There was no statistically significant difference in the number of patients who received antibiotics on discharge from ED between the pre- and post-RPP study periods.

## Data Availability

All data are contained within this article.

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
