# Peer review of "Diagnostic Stewardship—The Impact of Rapid Diagnostic Testing for Paediatric Respiratory Presentations in the Emergency Setting: A Systematic Review"

_children, 2022, doi:10.3390/children9081226_

Round 1
Reviewer 1 Report
Resistance to antibiotics is a severe public health concern. In clinical practice, the signs and symptoms of respiratory infections may be complicated and perplexing, especially in children; this can lead to misdiagnosis and the inappropriate prescription of antibiotics if the treating physician lacks competence. Some viral respiratory infections may be misdiagnosed as bacterial illnesses, which raises the need for needless antibiotic prescriptions and prolongs therapy. Rapid diagnostic tests may increase the precision of diagnosis, the specificity of post-diagnostic therapies, the usage of antibiotics, and the expense of follow-up care. The purpose of this systematic review is to determine if quick diagnostic tests for respiratory symptoms in paediatric emergency care influence the prescription, dosage, and duration of antibiotics, with practical implications for the management of antimicrobials. The manuscript is well-written and organized, including all required component for systematic review.
Major comments:
1. After a certain fast diagnostic test has been performed and a disease has been confirmed or ruled out, the quantity or kind of antibiotics used will naturally decrease; nonetheless, which rapid test should be performed in an emergency?
2. It was also noted that the median time from sample collection to result was 1 hour 52 minutes with the BioRP2 compared to 26 hours with IFA, while other rapid tests do not mention the time used; and hence, will waiting for the rapid test results before deciding on antibiotic medication delay the optimal treatment time?
3. The rapid test always cost a lot, especially for a full panel of different pathogens. If applied to all of the patients, the cost will may be much more than that for use of antibiotics. Please add this information or discuss.
Minor comments:
1. The inappropriate use of antibiotics, may lead to antibiotics resistance and also antibiotics’ side effects or induced bacterial pathogenesis, please add to the introduction.
2. In line 78, the ‘Are’ should be ‘Does’, Please check the rest of the text.
3. In line 350, ‘infants<28 years old’ should be ‘infants<28 days.
Author Response
We thank the reviewer for providing insightful comments on the manuscript. Please find our response to these comments below:
Major comments:
- After a certain fast diagnostic test has been performed and a disease has been confirmed or ruled out, the quantity or kind of antibiotics used will naturally decrease; nonetheless, which rapid test should be performed in an emergency?
Response: Thank you for bringing this to our attention, we have added further explanation into the paragraph in the discussion which explores the differences in time taken for RDTs, and included the fastest RDT with the widest panel as the ideal RDT in an emergency.
The added text reads as “Moreover, the ability for RDTs to provide sufficient diagnosis in an emergency setting should be explored. Currently, the BioRP2 RDT may be considered in an emergency as it has the ability to provide a full panel of different pathogens within 45 minutes [22].”
- It was also noted that the median time from sample collection to result was 1 hour 52 minutes with the BioRP2 compared to 26 hours with IFA, while other rapid tests do not mention the time used; and hence, will waiting for the rapid test results before deciding on antibiotic medication delay the optimal treatment time?
Response: Thank you for your comment. We have added a line into the discussion highlighting that despite the introduction of RDT’s, the delayed optimal treatment time may still result in the dispensing of empirical antibiotics.
This text reads as “It should be noted that the waiting time for the rapid test results may delay optimal treatment time, therefore the dispensing of empirical antibiotics may still be required.”
- The rapid test always cost a lot, especially for a full panel of different pathogens. If applied to all of the patients, the cost will may be much more than that for use of antibiotics. Please add this information or discuss.
Response: Thank you for your comment. The use of RDTs in all patients is deemed too expensive, and therefore the use of RDTs is recommended in only high-risk populations, as well as only during peak influenza seasons. This is also included in the Discussion as “In order to balance cost and benefit of RDTs, several studies suggested the use of RDTs in only high-risk populations or only during peak influenza seasons to ensure accurate diagnosis and reduce antibiotic use [14, 19].”
Minor comments:
- The inappropriate use of antibiotics, may lead to antibiotics resistance and also antibiotics’ side effects or induced bacterial pathogenesis, please add to the introduction.
Response: Thank you for your comment. This has now been addressed in the Introduction as “These inappropriate antibiotic prescribing behaviours are also associated with increased risk of adverse side effects of antimicrobial agents as well as increased medicalisation of typically self-limiting presentations [3]”
- In line 78, the ‘Are’ should be ‘Does’, Please check the rest of the text.
Response: Thank you. This has been corrected.
- In line 350, ‘infants<28 years old’ should be ‘infants<28 days.
Response: Thank you. This has now been corrected.
Reviewer 2 Report
Comments for Authors:
The Authors conducted a comprehensive systematic review about rapid tests for children with suspected respiratory infections. The methods appear sound and are clearly described. The data is nicely summarized and presented. My biggest concern are the conclusions that can be drawn from studies that mainly investigated antibiotic prescription rates but rarely clinical outcomes.
· You summarized the available literature from the past 20 years, without considering the time of the conduct of the respective studies. Have there been trends or new technologies that might have improved in the past 20 years?
· Most of the studies focused on antibiotic prescription rates. This is understandable for the individual studies, because choosing this study outcome increases the probability of a positive study result. However, it is very likely that negative rapid tests lead to less antibiotic prescription. In the absence of negative tests (or positive viral antigen tests), clinicians may prescribe antibiotics to be ‘on the safe side’. Showing lower rates of antibiotic prescriptions alone does not clinical benefit for patients. At least, non-inferiority of clinical outcomes has to be shown, to safely recommend rapid tests that lower antibiotic prescription rates. I think this aspect has to be addressed in the manuscript. You partially addressed this in the Discussion, but the absence of significant differences does not prove non-inferiority (these trials were usually not powered for clinically significant differences). Maybe you can find a little bit more clinical outcome data. Otherwise, these uncertainties should be mentioned.
Author Response
We thank the reviewer for providing insightful comments on the manuscript. Please find our response to these comments below:
- You summarized the available literature from the past 20 years, without considering the time of the conduct of the respective studies. Have there been trends or new technologies that might have improved in the past 20 years?
Response: Thank you for your comment. The included studies in the review cover a range of study periods ranging from November 1998 – November 2021 and there was no clear trend highlighted over the 20 years. This may be due to the differing study populations and study designs, as well as the use of differing RDT methods.
- Most of the studies focused on antibiotic prescription rates. This is understandable for the individual studies, because choosing this study outcome increases the probability of a positive study result. However, it is very likely that negative rapid tests lead to less antibiotic prescription. In the absence of negative tests (or positive viral antigen tests), clinicians may prescribe antibiotics to be ‘on the safe side’. Showing lower rates of antibiotic prescriptions alone does not clinical benefit for patients. At least, non-inferiority of clinical outcomes has to be shown, to safely recommend rapid tests that lower antibiotic prescription rates. I think this aspect has to be addressed in the manuscript. You partially addressed this in the Discussion, but the absence of significant differences does not prove non-inferiority (these trials were usually not powered for clinically significant differences). Maybe you can find a little bit more clinical outcome data. Otherwise, these uncertainties should be mentioned.
Response: Thank you for bringing this to our attention. we have added a section in the patient outcomes subsection within the discussion, as well as the conclusion, mentioning the uncertainty surrounding this issue and need for further research.
The added text in the Discussion section reads as “This is of concern, as the absence of a significant difference in patient outcomes between groups assessed with RDT as opposed to standard measures, does not represent non-inferiority. Therefore, further research needs to assess the implementation of RDTs powered for patient outcome measures, rather than antibiotic prescription rates alone. This will show RDTs efficacy for AMR as well as clinical outcomes, prior to imple-mentation in emergency settings”
The added text in the Conclusion section reads as “Further research is required to assess the effect of RDT implementation with a focus on patient clinical outcomes.”
Round 2
Reviewer 2 Report
Revised version is fine. No further comments from my side.